# Supercooled erythritol for high-performance seasonal thermal energy storage

**Sheng Yang** [1,2], **Hong-Yi Shi** [1,2], **Jia Liu** [1,2], **Yang-Yan Lai** [1,2], **Özgür Bayer** [3] & **Li-Wu Fan** [1,2] ✉

Seasonal storage of solar thermal energy through supercooled phase change materials (PCM) offers a promising solution for decarbonizing space and water heating in winter. Despite the high energy density and adaptability, natural PCMs often lack the necessary supercooling for stable, long-term storage. Leveraging erythritol, a sustainable mid-temperature PCM with high latent heat, we introduce a straightforward method to stabilize its supercooling by incorporating carrageenan (CG), a bio-derived food thickener. By improving the solid-liquid interfacial energy with the addition of CG the latent heat of erythritol can be effectively locked at a very low temperature. We show that the composite PCM can sustain an ultrastable supercooled state below −30 °C, which guarantees no accidental loss of the latent heat in severe cold regions on Earth. We further demonstrate that the common ultrasonication method can be used as the key to unlocking the latent heat stored in the CG-thickened erythritol, showing its great potential to serve as a high-performance, eco-friendly PCM for long-term seasonal solar energy storage.

Towards a carbon-neutral future, it is crucial to develop decarbonized space and water heating systems[1–4]. Space and water heating in winter, which accounts for ~60% of the energy consumption in buildings, is a major concern for life support and thermal comfort[5–10]. Solar energy has long been exploited as a sustainable resource, as solar photovoltaic power accounts for 4.66% of global electricity generation[11–13]. For energy-efficient heating or cooling in buildings, utilizing solar thermal energy in households is an alternative option as it eliminates the need to convert solar energy into electricity before utilizing it for heating purposes[14–17]. Additionally, solar thermal energy exhibits higher conversion efficiency compared to photovoltaics, resulting in a substantial improvement in the overall utilization efficiency of solar energy[18]. However, a significant technical obstacle arises when harvesting solar thermal energy for space heating, due to the temporal mismatch between abundant solar energy in summer and the high heating demands in winter[16,19–22]. This mismatch is particularly evident in some climate regions, e.g., in northwestern China and eastern Turkey. The solar irradiance can reach 2,200 MJ·m$^{-2}$ in summer with

extreme ambient temperatures above 40°C, whereas in winter, the temperature may plummet down to −30 °C with a low solar irradiance of only 990 MJ·m$^{-2}$[23,24].

Seasonal thermal energy storage (TES) has been utilized to mitigate this mismatch by storing excessive solar energy in summer and releasing it for space and water heating in winter when needed[9,25–36], as illustrated in Fig. 1a. Seasonal TES has served as a low-carbon heating solution for both the above-mentioned regions and northern European countries, where the annual energy consumption is as high as 4,063 kWh per person for space and water heating[37]. However, the significant and unavoidable heat loss poses a crucial challenge for implementing seasonal TES technologies over long-term storage periods[38–42], in analogy to the self-discharge issue associated with rechargeable batteries[43]. The temperature drop in seasonal TES systems due to heat loss, which is an irreversible thermodynamic process, reduces considerably the "Coulombic efficiency" of such "thermal batteries" for seasonal solar heating[44]. For example, in existing seasonal sensible heat storage projects, which utilize water or soil as the TES

[1]State Key Laboratory of Clean Energy Utilization, Zhejiang University, Hangzhou, People's Republic of China. [2]Institute of Thermal Science and Power Systems, School of Energy Engineering, Zhejiang University, Hangzhou, People's Republic of China. [3]Department of Mechanical Engineering, Middle East Technical University, Ankara, Türkiye. ✉e-mail: liwufan@zju.edu.cn

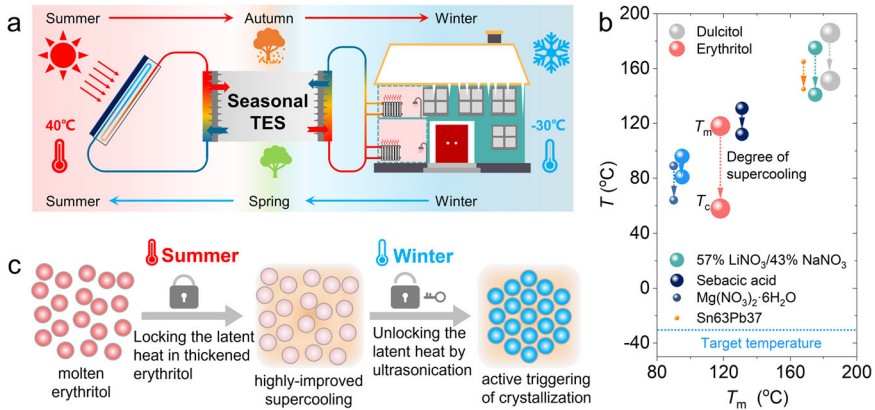

**Fig. 1 | A seasonal solar TES system using thickened-erythritol with ultrastable supercooling. a** Concept of storing solar thermal energy in summer for space and water heating in winter by seasonal thermal energy storage (TES). **b** Comparison between erythritol and other PCMs with high degrees of supercooling over the medium temperature range from 80°C to 200°C, where the size of the symbol represents the relative magnitude of the latent heat of fusion. **c** Illustration of the idea of improving the degree of supercooling of erythritol by adding a thickener and triggering the discharging process by an external mechanism like ultrasonication.

material, significant heat losses were observed to be up to 60% during long-term storage periods[45,46]. Utilizing expensive ultra-insulation materials can only decelerate the heat loss process, but will lead to a substantial increase in costs for household distributed TES systems.

To this end, supercooled phase change materials (PCMs) provide an option for seasonal TES with much less thermal insulation concern and higher thermal storage density[47]. A PCM can remain at the supercooled liquid state when the ambient temperature drops down to way below its nominal melting point ($T_m$), but releasing the latent heat at its melting point after the crystallization process being triggered[48]. In other words, the heat (harvested from the sun in summer) can be "locked" in a supercooled PCM when the weather transits to winter, thus the strong heat loss over the several months will not cause accidental releasing of the latent heat stored[49–53]. In addition, due to the stable supercooling behavior, supercooled PCMs can also provide options for challenging environments, such as spacecraft, mobile heating, polar buildings, etc.

The operating temperatures of domestic heating systems (~80°C) and household solar thermal collectors (<200°C) necessitate that the melting point of an appropriate PCM should be within this medium temperature range (80−200°C). As shown in Fig. 1b, several PCM candidates are available over this temperature range[54–58], including erythritol ($T_m = 118$°C), MgCl$_2$·6H$_2$O ($T_m = 118$°C), adipic acid ($T_m = 152$°C), dulcitol ($T_m = 186$°C). All these candidate PCMs possess a considerable degree of supercooling ($T_{sup} = T_m − T_c$, with $T_c$ representing the crystallization point), and erythritol appears to be the most promising one. In particular, erythritol belongs to the family of sugar alcohols, which has long been used in the food and pharmaceutical industry as a sugar substitute[59]. Among all the above-mentioned candidate PCMs in the medium temperature range, erythritol not only holds the highest degree of supercooling ($T_{sup} = $~60°C)[48,54,60,61] and the highest latent heat of fusion ($H_m = $~340 J·g$^{-1}$)[62–67], but also exhibits the greatest sustainability because it is manufactured by biomaterials like glucose[68].

However, the relatively high degree of supercooling of erythritol still seems to be insufficient for seasonal TES applications. As mentioned, some climate regions, especially the monsoon climate regions, feature scorching summers and severe winters, requiring the PCM to perform under extremely cold conditions, e.g., being able to remain in a supercooled liquid state down to −30°C or even lower. This situation poses a great challenge for all existing natural PCMs, making it imperative to develop better PCMs that exhibit ultrastable supercooling behavior[69,70]. Therefore, erythritol is deemed to be the most promising starting point for developing such an ideal seasonal PCM. In recent literature, several studies have reported results on stabilizing the supercooling behavior of erythritol by introducing additives. For example, Puupponen and Turunen et al.[71,72] dispersed erythritol in sodium polyacrylate matrix for long-term TES, successfully improving the degree of supercooling to about 110°C and stabilizing it for up to 97 days. More recently, Li et al.[73] added alkali hydroxides into erythritol to increase the activation energy barrier for solidification, leading to a stabilized supercooled state at room temperature, i.e., $T_{sup} = $-100°C, for 30 days. However, the low-temperature stability requirement is equivalent to increasing the degree of supercooling of erythritol to an ultrahigh value of >150°C. To date, there is no approach for improving the supercooling behavior of erythritol to such an ultrastable level.

Sugar alcohols, including erythritol, have been characterized with high viscosities[74], which is believed to be relevant to their remarkable supercooling effect because a high viscosity lowers both the molecular mobility and crystal growth rate in the supercooled phase[75,76]. The essence of improving the degree of supercooling of a PCM is to inhibit the incipience and development of crystallization. As illustrated in Fig. 1c, this fact inspires us to seek a highly improved stability of supercooled erythritol by simply making it more viscous with the addition of a thickening reagent.

In this work, we tested a variety of common thickeners and studied their effectiveness in improving the supercooling behavior of erythritol. As the thickener-enhanced erythritol poses a consequent issue that the latent heat stored is very well protected from being discharged, we further identified the ultrasonication method as a "key" to unlock the latent heat after being stored in the additive-thickened, highly supercooled erythritol for a long period, thus enabling a controllable way of releasing the latent heat when needed.

## Results

### Chemical and thermal characterization on thickened erythritol
In this work, we tested several common cost-effective thickeners, including three eco-friendly, bio-derived food thickeners, i.e., carrageenan gum (CG), guar gum (GG), xanthan gum (XG). Also tested are other common thickeners of polyvinyl alcohol (PVA), carboxymethyl cellulose (CMC) and sodium carboxymethyl cellulose (CMCNa). We prepared the composite PCM samples by grinding and melting the thickeners at various loadings with erythritol. After a comprehensive comparison, we found that CG performs the best in improving the degree of supercooling of erythritol, so the following discussion will only focus on the results of CG. Detailed comparative results among

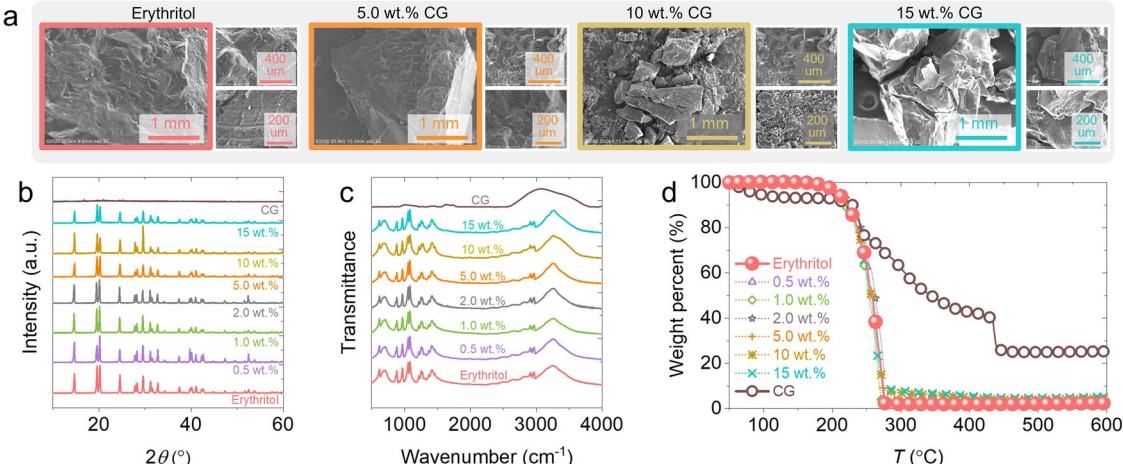

**Fig. 2 | The structure and thermal stability of erythritol at different CG loadings. a** The scanning electron microscope (SEM) images on the microscale morphology of the composite erythritol samples with 0 wt.%, 5 wt.%, 10 wt.%, and 15 wt.% of CG. **b** The X-ray diffraction (XRD) pattern, **c** Fourier transform infrared (FTIR) identification of hydrogen bonds and -OH groups, and **d** thermal gravimetric analyzer (TGA) testing of the CG-thickened erythritol samples.

the gum-type thickeners and other common thickeners are presented in Supporting information (SI 1, 2, and 4).

The addition of CG into erythritol leads to a softening transformation in its structure. The scanning electron microscope (SEM) images in Fig. 2a show that the surface structure of pure erythritol is smooth and uniform. However, the inclusion of CG introduces fragmentation and porosity to the crystal structure of the erythritol-CG composites. Moreover, when the CG content surpasses 10 wt.%, pore, and particle structures become noticeable. This phenomenon occurs because CG, a polysaccharide substance, exhibits thickening and gelling properties. The interaction between CG and erythritol involves hydrogen bonds and electrostatic forces among erythritol molecules. Consequently, these interactions cause the connection between erythritol molecules to relax, resulting in an overall increase in softness. The relaxation of connections between erythritol molecules, as indicated in Fig. 2a, leads to an increase in the overall softness of the structures.

While the addition of CG modifies the overall structure of erythritol as a PCM, it does not affect its minimum crystal structure. As a result, the peaks of erythritol in the X-ray diffraction (XRD) spectrum remain unaffected by the presence of CG, as confirmed in Fig. 2b. The diffraction peaks of erythritol are in excellent agreement with the standard patterns as well as the results reported in previous study[77]. It is also clear that the diffraction peaks of the CG-thickened erythritol represent the superposition of the patterns of erythritol and CG, suggesting that no new compounds were created during our preparation.

The hydroxyl functional group on each carbon atom results in strong hydrogen bonding between neighboring erythritol molecules. This force not only enables erythritol to absorb a great amount of heat during melting, serving as the source for high heat storage density, but also restricts the mobility of erythritol molecules in a molten state, leading to high intrinsic viscosity. Materials that can form intermolecular hydrogen bonds appear, in general, to be more viscous. In addition, the interfacial energy is related to this force. The surface molecules are attracted by the internal molecules of the liquid, and are pulled toward the liquid inside, so the surface tends to contract actively and hence to have a higher interfacial energy[78]. As shown in Fig. 2c, the Fourier transform infrared (FTIR) results reveal that the addition of CG does not disrupt the hydroxyl functional groups, as well as the hydrogen bond network among erythritol molecules. At a high loading of CG (15 wt.%), stable primary (wavenumber = 1053.9 cm$^{-1}$) and secondary (wavenumber = 1080.9 cm$^{-1}$) -C-OH groups of erythritol

as well as strong hydrogen bond interactions (wavenumber between 3000−3300 cm$^{-1}$) can be found. At this high loading, the hydrogen bonding peaks even become stronger, as can be seen in Fig. 2c.

Thermal stability refers to the ability of a PCM to withstand high temperatures during the charging process, especially in case of unexpected off-design temperature rises. The thermal gravimetric analyzer (TGA) curve in Fig. 2d demonstrates the excellent thermal stability of erythritol, exhibiting weight loss starting from the temperature of 180°C. The thermal stability of CG-thickened erythritol at all loadings tested remains consistent with that of pure erythritol. Despite the relatively poor thermal stability of CG (subject to thermal decomposition above 60°C), the addition of such minute amounts (up to 15 wt.%.) of CG does not seem to disturb the thermal stability of erythritol. Only a mass loss of ~7.8% is observed when the temperature is up to 220°C, allowing the CG-thickened erythritol to work well at the common operating temperatures of household solar thermal collectors.

## Highly-improved supercooling behavior of CG-thickened erythritol

In order to achieve long-term seasonal TES in extremely cold environments, an ultrahigh degree of supercooling is desired. The supercooling behavior of pure erythritol and CG-thickened erythritol under non-isothermal condition was tested by differential scanning calorimetry (DSC) with charging and discharging cycles over the temperature range between 170°C and −100°C, at a ramping rate of 10 K·min$^{-1}$. As shown in Fig. 3a-i, pure erythritol fails to maintain a supercooled state upon cooling, resulting in an undesired release of the latent heat (~198 J·g$^{-1}$) at the temperature of 33°C. This observation confirms that the low degree of supercooling of pure erythritol is insufficient to meet the requirements for seasonal TES in the above-mentioned regions featuring severe cold winter, down to −30°C or even lower. As what we looked for, the CG-thickened erythritol exhibits much higher degrees of supercooling. As illustrated in Fig. 3a-ii, the 5 wt.% CG-thickened erythritol starts to crystallize until the temperature decreases to 16°C, and releases the latent heat of 140 J·g$^{-1}$. With increasing CG loading, the supercooling behavior becomes more stable. Especially, the two most concentrated samples (10 wt.% and 15 wt.%) could maintain an ultrastable supercooled state in a severely cold environment, even when the temperature drops down to around −50°C until the supercooled sample becomes vitrificated[69], as indicated by the inflection on the discharging curve of the 15 wt.% case in Fig. 3a-iv.

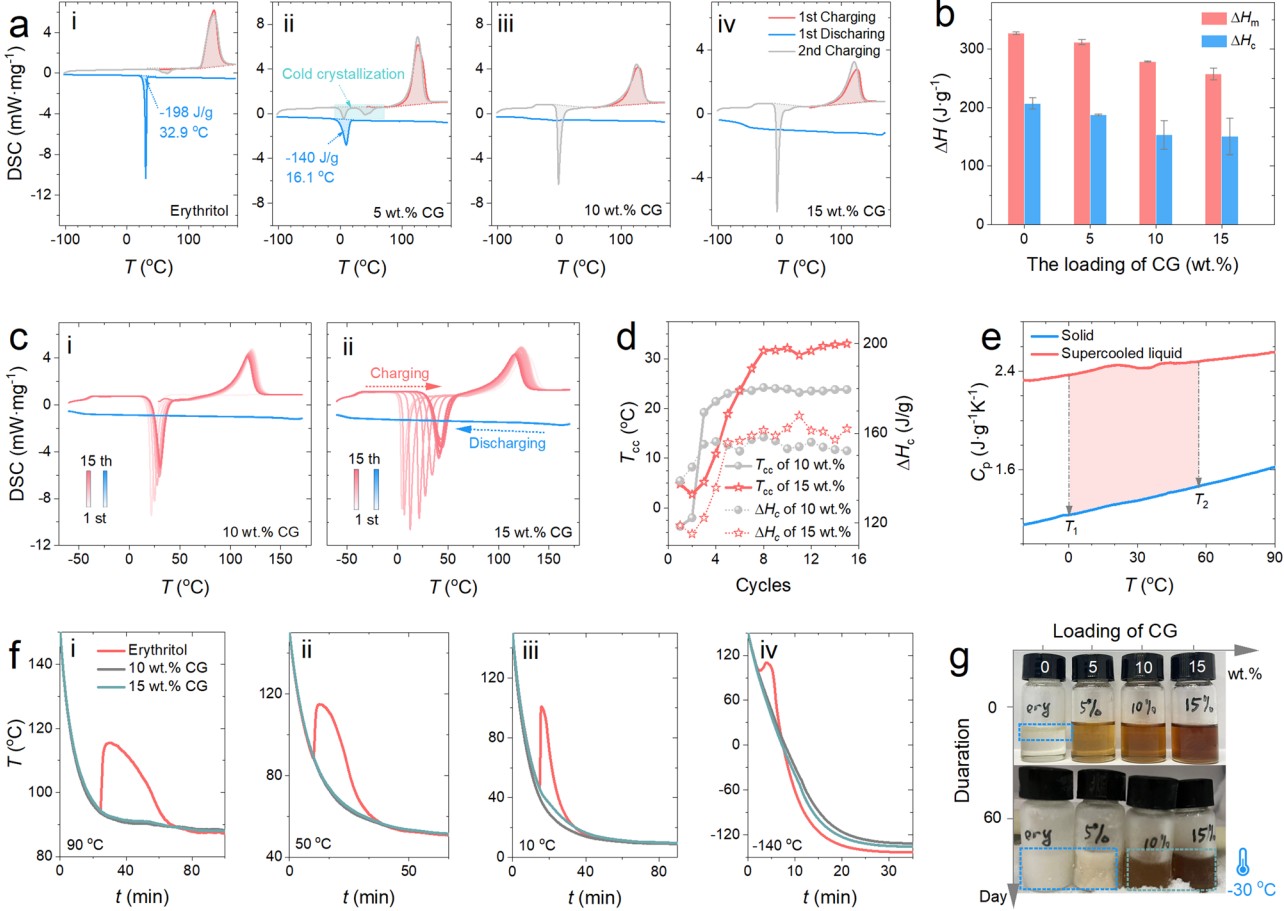

**Fig. 3 | The supercooling and cycling stability of erythritol at different CG loadings. a** The supercooling behavior and cold crystallization of erythritol and CG-thickened erythritol under non-isothermal test conditions. **b** The latent heat of fusion ($\Delta H_m$) and crystallization ($\Delta H_c$) values of erythritol and CG-thickened erythritol. **c** The cycling stability of the two most concentrated CG-erythritol samples after 15 charging and discharging cycles over the temperature range from −50°C to 150°C. **d** The change in the temperature of cold crystallization ($T_{cc}$) and $\Delta H_c$ of the 10 wt. % and 15 wt.% CG thickened erythritol with an increase of cycles. **e** Comparison of the specific heat capacity of erythritol between solid and liquid states. **f** The supercooling behavior of erythritol and CG-thickened erythritol under isothermal temperatures of 90 °C, 50 °C, 10 °C, and −140 °C. **g** The photos of erythritol and CG-thickened erythritol samples maintained in a cold environment at the temperature of −30 °C after 60 days.

In addition, there is a serious cold crystallization phenomenon that can be observed during the subsequent charging process, as indicated by the exothermic peak on the second charging curves in Fig. 3a. The occurrence of cold crystallization, which is often observed in polymers, stems from the fact that only part of the erythritol molecules crystallizes during the first discharging process, resulting in a partial amorphous state. Then the remaining non-crystallized part of erythritol molecules can be triggered to crystallize and release the remaining latent heat (of crystallization) upon being heated during the subsequent charging process. As can be seen in Fig. 3a, increasing the loading of CG intensifies the cold crystallization phenomenon. For example, at the loading of 5 wt.%, the latent heat of ~38.9 J·g⁻¹ is released during the 2nd charging process. Furthermore, when the loading increases to above 10 wt.%, the CG-thickened erythritol can no longer crystallize upon cooling down to below −100°C and all of the latent heat of crystallization is released during the 2nd charging process.

As shown in Fig. 3b, the addition of CG leads to a reduction in both the $\Delta H_m$ and $\Delta H_c$ of erythritol, and the decreasing trends are almost linear with increasing the CG loading. Due to the partial crystallization effect and partial release of the heat in the form of sensible heat during the cooling process, $\Delta H_c$ is always lower than $\Delta H_m$ at any loadings (see Supporting information (SI 1)). Compared to pure erythritol ($\Delta H_m = 327.3 \pm 2.0$ J·g⁻¹, $\Delta H_c = 207.0 \pm 4.5$ J·g⁻¹), the most concentrated

15 wt.% CG-thickened erythritol still possess fairly high latent heat values ($\Delta H_m = 259.3 \pm 15.5$ J·g⁻¹, $\Delta H_c = 150.1 \pm 31.3$ J·g⁻¹). Note that the averaged $\Delta H_c$ of the 15 wt.% CG-thickened erythritol is only slightly lower than that of the 10 wt.% sample ($153.6 \pm 26.5$ J·g⁻¹), showing a nonlinear decreasing trend at high loadings.

As presented in Fig. 3c, this nonlinear variation is related to the difference in cold crystallization behavior upon consecutive charging and discharging cycles between the 10 wt.% and 15 wt.% samples. On the charging curves for both the 10 wt.% and 15 wt.% samples, the peak representing cold crystallization keeps shifting to the right, i.e., the temperature of cold crystallization ($T_{cc}$) becomes higher, as the cycling proceeds, and the extent of shifting for the 15 wt.% case is more remarkable, as clearly shown by the variations of both $T_{cc}$ and $\Delta H_c$ plotted in Fig. 3d over the 15 cycles. The greater increase of $\Delta H_c$ of the 15 wt.% sample upon cycling makes it surpass the value of the 10 wt.% sample after the 8th cycle, leading to the very close averaged $\Delta H_c$ values (over the several cycles) between these two cases in Fig. 3b. Not surprisingly, due to the larger variation range, the error bar is also longer for the 15 wt.% sample.

Recall that for the samples with high CG loadings, the latent heat of crystallization totally stems from cold crystallization. Therefore, the origin for this unusual increase of $\Delta H_c$ of the 15 wt.% sample with cycling is attributed to the discrepancy of the specific heat capacity between the solid (crystallized) and supercool liquid phases of

erythritol (and erythritol-based composite PCM). As shown in Fig. 3e, the specific heat capacity has two different values over the temperature range where the ultrastable supercooled state can be held. At the same temperature in this range, the specific heat capacity of erythritol in the supercooled liquid phase is up to 2.6 J·g$^{-1}$K$^{-1}$, which is much greater than the -1.6 J·g$^{-1}$K$^{-1}$ in the solid phase. For example, compared to the lower temperature $T_1$, the cold crystallization occurs at $T_2$ will be able to release more heat, as represented by the shaded area marked in Fig. 3e. Details concerning the effect of specific heat capacity on the latent heat of crystallization is shown in Supporting information (SI 1).

Figure 3c also exhibits the good cycling performance of the CG-thickened erythritol. Over the course of 15 cycles, the CG-thickened erythritol consistently maintains its supercooled state during the cooling process. After 15 cycles, the $\Delta H_m$ of the 10 wt.% and 15 wt.% samples decrease only by 10.4% and 11.0%, respectively. In practice, a seasonal PCM will undergo annual cycles of charging and discharging. Such slight degradation of the latent heat storage capacity over 15 cycles suggests that the CG-thickened erythritol can serve as a high-performance PCM for seasonal solar energy storage for more than 15 years.

In addition, it is noted that the degree of supercooling differs between isothermal and non-isothermal cooling processes. For the same PCM, a lower value of the degree of supercooling is usually observed under isothermal conditions. Therefore, isothermal tests were also performed on the pure and composite erythritol samples, with details being provided in Supporting information (SI 1). As shown in Fig. 3f, pure erythritol crystallizes (i.e., discharges the latent heat) at a temperature of 94.7°C, which is much higher than that observed under non-isothermal conditions. The two CG-thickened erythritol samples with high loadings can easily maintain a stable supercooled state at an isothermal temperature of 90°C without releasing latent heat. When the isothermal temperature drops down to 60°C, pure erythritol is also not able to maintain the supercooled state. In contrast, the CG-thickened erythritol exhibits an ultrastable supercooling behavior, even at temperatures down to below −100°C, during the isothermal cooling process, similar to the performance observed in DSC tests. As shown in Fig. 3f, the CG-thickened erythritol demonstrates the ability to maintain a supercooled state at very low temperatures, confirming again its great potential for long-term TES of solar energy in severe cold regions.

The difference in supercooling behavior between pure erythritol and CG-thickened erythritol was also visually observed in Fig. 3g. When taking the photos at room temperature (at day 0), the molten erythritol in the small bottle starts to crystallize from the top surface, whereas the CG-thickened erythritol samples are all in a supercooled liquid state. After moving to a cold chamber set at −30°C, the pure erythritol and 5 wt.% sample can crystallize within one day, whereas the two heavily thickened 10 wt.% and 15 wt.% samples successfully maintain an ultrastable supercooled state without freezing for more than 60 days.

## Rheological behavior in the supercooled state of thickened erythritol

To support the analysis on the basis of viscosity growth and beyond, we characterized the rheological behavior of the pure and CG-thickened erythritol samples using a high-precision rotational rheometer. Details about the rheological measurement are provided in Supporting information (SI 1). A small organic compound with a linear molecular structure, erythritol behaves as a Newtonian fluid in its molten state. Figure 4a demonstrates the linear relationship between shear stress and shear rate over a wide range from 1 s$^{-1}$ to 10,000 s$^{-1}$. Moreover, the shear stress of erythritol increases significantly by 2.8 times when the temperature decreases from 150°C to 120°C. With the addition of CG, the shear stress of the CG-thickened erythritol becomes much greater than that of pure erythritol at the same shear

rate and temperature. For example, at a shear rate of 10,000 s$^{-1}$ and at 120°C, compared to pure erythritol, the shear stress of 10 wt.% CG-thickened erythritol increases by nearly 70% to 828 Pa, and the shear stress of 15 wt.% CG-thickened erythritol grows by more than 160% to 1277 Pa.

At low shear rates (0.001 s$^{-1}$ to 1 s$^{-1}$), the viscosity of pure erythritol decreases at first and then reaches a constant value at ~1 s$^{-1}$, as shown in Fig. 4b. However, with increasing the CG loading, the viscosity is improved significantly, leading to a gradual rheological behavior transition from Newtonian to non-Newtonian. When adding CG at 10 wt.%, the regime of Newtonian fluid disappears, causing the CG-thickened erythritol to exhibit shear-thinning behavior over the range of shear rate from 1 s$^{-1}$ to 10,000 s$^{-1}$. Approaching the shear rate of 1 s$^{-1}$, the change in viscosity decreases with increasing the shear rate, as shown in Fig. 4b.

When the loading becomes 15 wt.%, the CG-thickened erythritol exhibits a Newtonian fluid regime within the range of shear rate between 0.001 s$^{-1}$ and 1 s$^{-1}$, and a shear-thinning behavior between 0.1 s$^{-1}$ and 10,000 s$^{-1}$. Note that making accurate measurements of the shear stress and viscosity at low shear rates (0.001 s$^{-1}$ to 1 s$^{-1}$) is challenging, thus generating some errors in the results. Clearly, the addition of CG causes significant changes in the rheological behavior of erythritol. Figure 4c depicts the viscosity testing results of erythritol with 0.5 wt.% and 5 wt.% CG additions, showing that a minute amount of only 0.5 wt.% CG can increase the viscosity of erythritol by 3.2 times. A 5 wt.% CG addition boosts the viscosity by 20.6 times, and the most concentrated 15 wt.% CG addition leads to a significant 314.7-fold increase.

## Mechanisms of thickening on the stabilized supercooling behavior

As discussed, the inclusion of CG has a significant impact on the degree of supercooling of erythritol. The ultrahigh supercooling behavior of erythritol and other PCMs means the absence of a crystallization process, which consists of two steps, as shown in Fig. 5a. The first step involves the growth of embryos into nuclei with a critical nucleation radius ($R_c$), followed by the second step where the nuclei develop into complete crystals. While it has been widely understood that the viscosity (μ) influences significantly the step for crystal growth, another less concerned parameter that plays a critical role in the initial step for nucleation is the solid-liquid interfacial energy ($\gamma_{sl}$) of the PCM.

In the first step, as shown in Fig. 5b, $\gamma_{sl}$ influences the value of Gibbs free energy $\Delta G^*_{ho}$, the critical size of nucleus $R_c$, and the rate of formation of nuclei $I^{homo}$, during the crystallization process, and the values of these three parameters together dictate the crystallization process and supercooled state of erythritol. In Fig. 5b, $r$ is the radius of embryos, $\Delta G_v = \rho \Delta s_f \Delta T$ is the volume term in the Gibbs free energy, and $\rho$ (kg·m$^{-3}$) is the density of the PCM, $\Delta s_f$ (J·g$^{-1}$K$^{-1}$) is the specific entropy of fusion which is the ratio of $\Delta H_m$ to the temperature range for melting, $\Delta T$ (K) is the temperature difference between the temperature of supercooled PCM and its melting point, and $k_B$ is the Boltzmann's constant (1.38×10$^{-23}$ J·K$^{-1}$)[79]. It is obvious that increasing the $\gamma_{sl}$ of erythritol results in a higher threshold of $\Delta G^*_{ho}$ and a larger $R_c$ of crystallization, as shown in Fig. 5c. This consequence significantly increases the difficulty of spontaneous crystallization, and hence the release of latent heat, for erythritol, indicating the achievement of a more stabilized supercooled state. Furthermore, a higher $\gamma_{sl}$ reduces $I^{homo}$, thereby making the growth from embryos to nuclei more challenging for erythritol[80,81]. Herein, improving the $\gamma_{sl}$ of erythritol, for raising the obstacles of forming critical nucleus and lowering the $I^{homo}$, is deemed to be an effective means of enhancing its degree of supercooling. The observed phenomenon is expressed as building a "thermodynamic dam", i.e., raising the energy barrier to nucleation and crystallization, to increase the degree of supercooling of erythritol, as shown in Fig. 5d. The highly-stabilized supercooling behavior can also

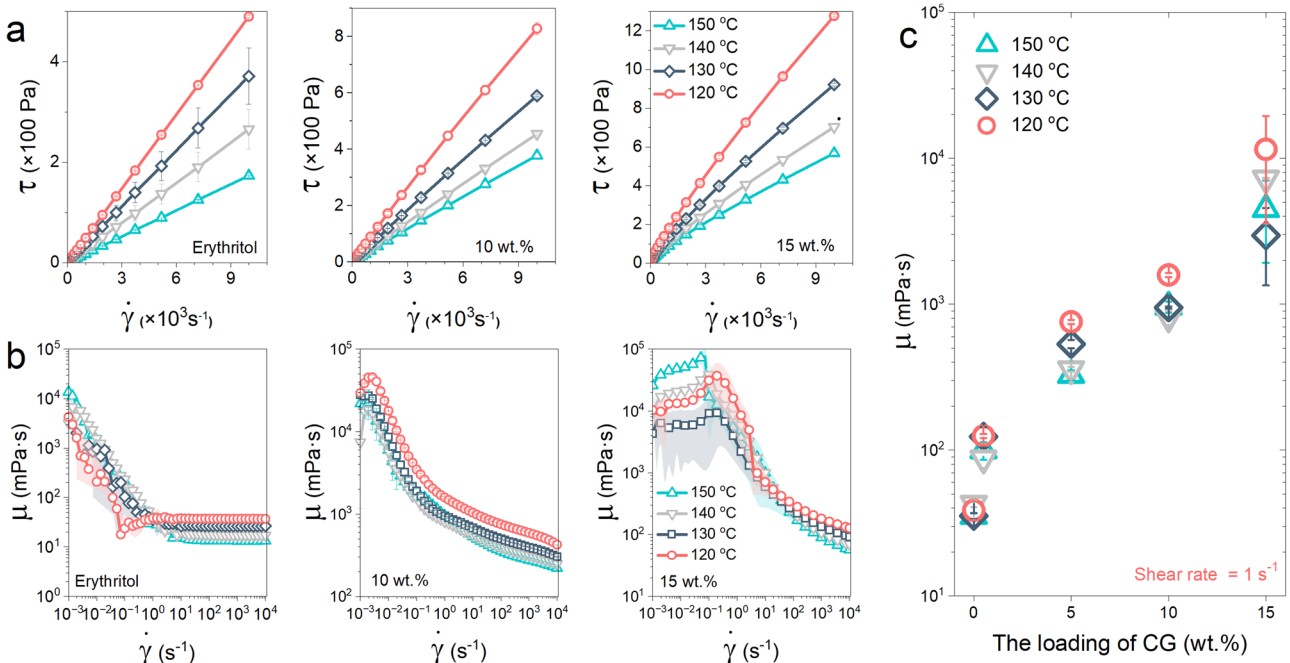

**Fig. 4 | The rheological behavior in supercooled state. a** The shear stress of erythritol and CG-thickened erythritol as a function of shear rate. **b** The dynamic viscosity of erythritol and CG-thickened erythritol as a function of shear rate. **c** The effects of CG on the dynamic viscosity of erythritol with various loadings at the shear rate of $1\,s^{-1}$.

address the thermal insulation and heat loss issue caused by accidental crystallization during long-term seasonal TES.

In addition, the well-protected latent heat stored in the highly-stabilized supercooled state of thicken erythritol poses a consequent challenge in releasing the latent heat when required. A passive triggering method is thus deemed to be necessary for timely release of the latent heat upon the discharging request in winter, like drilling a drainage path on the "dam", as shown in Fig. 5e. The triggering issue will be addressed later.

In order to provide a quantitative evaluation of the "thermodynamic dam" by addition of CG, we determined the interfacial properties between the solid and liquid phases by measuring the contact angle of a drop of supercooled liquid PCM sample (pure or CG-thickened erythritol) located on a smooth solid surface made of the same material in crystallized state. As shown in Fig. 5f, the contact angle $\theta$ is determined as the angle between the solid-liquid interface and the gas-liquid interface. We performed the measurements at 80°C because of the observation that the surface of solid erythritol acts as crystalline nuclei when a supercooled erythritol droplet is in contact with the surface, promoting nucleation and causing rapid solidification of the deposited droplet before reaching equilibrium at lower temperatures, such as 70°C. Therefore, such a relatively low surface temperature impedes the measurement of the contact angle under stable conditions. At higher temperatures, such as 90°C, the reduction in the melting point of the CG-thickened erythritol, as compared to pure erythritol, leads to melting and softening of the underlying solid surface, which, in turn, introduces undesirable difficulty in the measurement as well. As shown in Fig. 5f, the $\theta$ of pure erythritol is found to be 31.3°, and is increased by 72.2% to 53.9°, when 15 wt.% CG was added.

The relationship among the various interfacial tensions can be given by the classical Young's equation[76], which reads

$$\gamma_{sg} = \gamma_{sl} + \gamma_{lg}\cos\theta_{sl} \qquad (1)$$

where $\gamma_{sg}$, $\gamma_{lg}$, $\gamma_{sl}$ are the interfacial tensions of the interfaces between solid erythritol and air, supercooled liquid erythritol and air, supercooled liquid erythritol and solid erythritol, respectively. A manipulation of Eq. (1) leads to the evaluation of $\gamma_{sl}$, as given by

$$\gamma_{sl} = \frac{\gamma_{lg}}{2}\left(\sqrt{1+\sin^2\theta_{sl}} + \cos\theta_{sl}\right) \qquad (2)$$

As illustrated in Fig. 5f, $\gamma_{lg}$ can be measured using the pendant drop method, with the calculation process being given in Supporting information (SI 3). As shown in Fig. 5g, compared with pure erythritol ($\gamma_{sl} = 33.98 \pm 2.02$ mJ·m$^{-2}$), the $\gamma_{sl}$ of 15 wt.% CG-thickened erythritol at 80°C is increased by nearly 45% to $49.39 \pm 0.506$ mJ·m$^{-2}$. It means that, the required $\Delta G^*_{ho}$ increases by about 4.5-fold (see Fig. 5h) and the required $R_c$ increases by nearly twice of the CG-thickened erythritol (see Fig. 4i). More importantly, the $I^{homo}$ decreases by more than $10^{160}$ times at the temperature of 80°C (see Fig. 5j).

In the first step of crystallization, it is widely accepted that the concentration of embryos needs to reach $10^6$·m$^{-3}$ in the supercooled liquid to form nuclei. However, the significant enlargement of $R_c$ caused by the increased $\gamma_{sl}$ requires a much higher concentration of up to $10^{13}$·m$^{-3}$ of embryos in CG-thickened erythritol[79]. Therefore, the nucleation becomes more challenging. In addition, based on the equation of $I^{homo}$ in Fig. 5b, the relationship between $I^{homo}$ and temperature $T$ and $\gamma_{sl}$ can be obtained. As a function of temperature in Fig. 5k-i, the $I^{homo}$ of erythritol is significantly decreased by $10^{10}$ times with the addition of CG. When the temperature approaches 0 K or the melting point of erythritol (391 K), the drop in $I^{homo}$ becomes much sharper. Compared to pure erythritol, the $I^{homo}$ of the 15 wt.% CG-thickened erythritol decreases by more than $10^{160}$ times at the temperature of 353 K. At a constant temperature, the value of $I^{homo}$ decreases with increasing the $\gamma_{sl}$.

In theory, erythritol cannot crystallize spontaneously at a temperature higher than 100°C (373 K), as seen in Fig. 5k-ii, but our experimental results show that it fails to crystallize a temperature higher than 60°C (333 K)[48]. Because in the supercooled state, erythritol undergoes both crystal growth from embryos to nuclei and embryo decay. So, it requires a minimum $I^{homo}$ to cover the rate of embryo decay. It can be considered that the $I^{homo}$ of erythritol at 60°C (333 K), and with $\gamma_{sl}$ of 33.98 mJ·m$^{-2}$, is the minimum rate at which it can

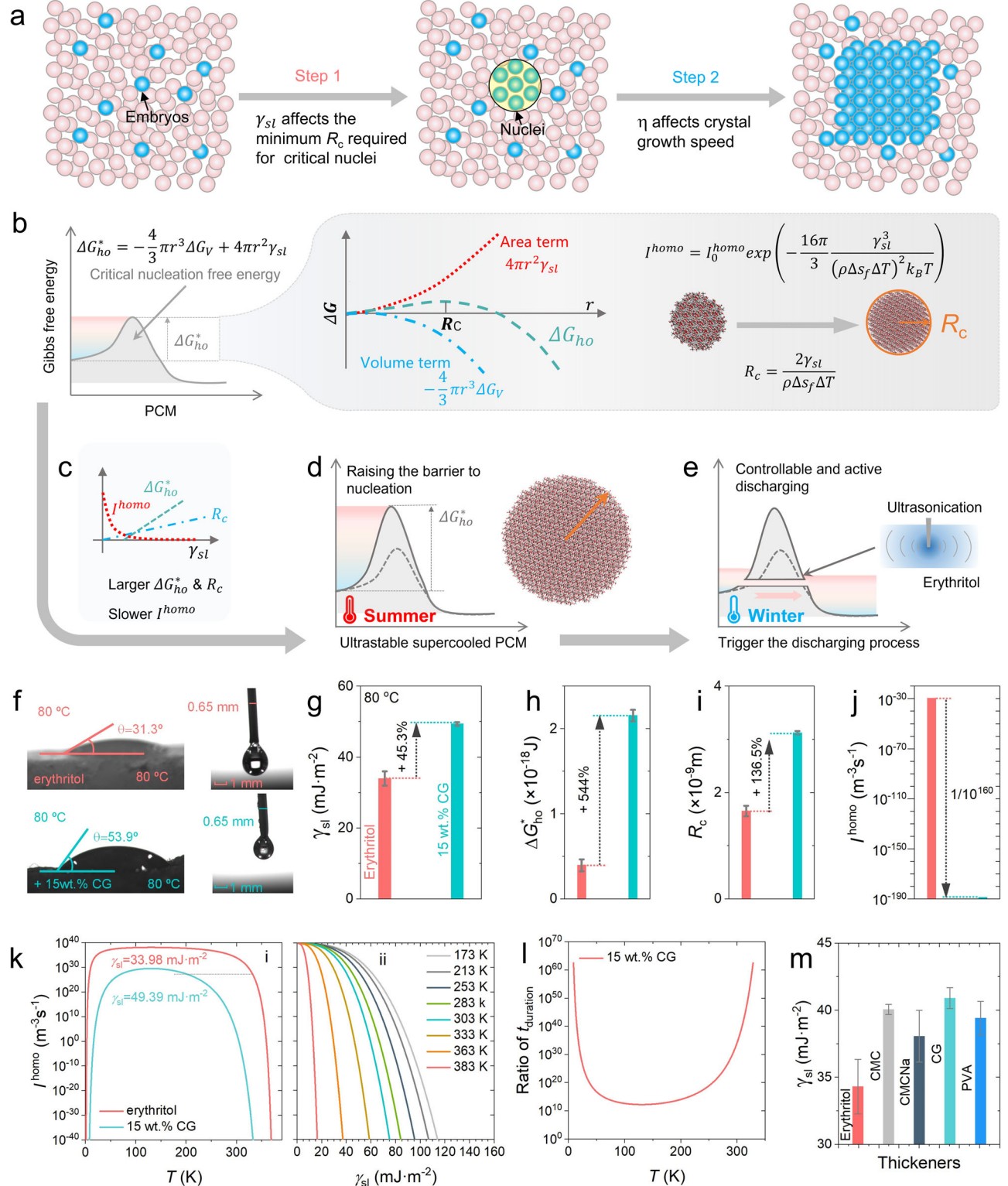

spontaneously crystallize. With the addition of CG, the theoretical minimum supercooling temperature can be lowered from 60°C to −70°C. The enlargement of $R_c$ and the slowing down of the $I^{homo}$ result in a more significant increase in the time needed to form complete nuclei from the embryo. As depicted in Fig. 5l, the duration of the nucleation process is extended by a factor of at least $10^{10}$.

Therefore, ultrastable supercooling behavior, with an ultrahigh degree of supercooling, of erythritol is achieved through the dual effects of thickening by CG, leading to increase in both the interfacial

energy and viscosity. As mentioned, we also tested other common types of thickeners, including PVA, CMC, and CMCNa, see in Supporting information (SI 4), and found that all of them perform worse than CG in stabilizing the supercooling behavior of erythritol. Despite both CMC and CMCNa exhibiting a more pronounced increase in viscosity, their lower effectiveness in enhancing the interfacial energy leads to an easier formation of crystal nuclei in the first step of crystallization, as shown in Fig. 5m. As a result, CG stands out as the best, among all the thickeners tested, in stabilizing the supercooling behavior of erythritol.

**Fig. 5 | The mechanisms of thickening on the stabilized supercooling behavior. a** The two steps of a typical crystallization process, where the first step involves the growth of embryos into nuclei with a critical nucleation radius, followed by the second step for the nuclei developing into complete crystals. **b** Illustration of the influence of Gibbs free energy $\Delta G^{*}_{ho}$, the critical size of nucleus $R_c$, and the rate of formation of nuclei $I^{homo}$ on the supercooling behavior, where $r$ is the radius of embryos, $\Delta G_v = \rho \Delta s_f \Delta T$ is the volume term in the Gibbs free energy, and $\rho$ (kg·m$^{-3}$) is the density of the PCM, $\Delta s_f$ (J·g$^{-1}$K$^{-1}$) is the specific entropy of fusion which is the ratio of $\Delta H_m$ to the temperature range for melting, $\Delta T$ (K) is the temperature difference between the temperature of supercooled PCM and its melting point, and $k_B$ is the Boltzmann's constant ($1.38 \times 10^{-23}$ J·K$^{-1}$)[79]. **c** Relationship between the $\gamma_{sl}$ and $\Delta G^{*}_{ho}$, $R_c$, and $I^{homo}$. **d** The concept of building a thermodynamic "dam" by raising the energy barrier to nucleation and crystallization to improve the degree of supercooling of erythritol. **e** Controllable triggering of the discharging process of

the CG-thickened erythritol by ultrasonication in a cold environment, like drilling a "drainage path" in the thermodynamic "dam". **f** Comparison of the contact angle $\theta_{sl}$ and surface tension, determined by sessile drop and pendant drop methods, respectively, between supercooled erythritol and CG-thickened erythritol (15 wt.%) at 80 °C. Comparison of the **g** interfacial energy $\gamma_{sl}$, **h** Gibbs free energy $\Delta G^{*}_{ho}$, **i** critical size of nucleus $R_c$, and **j** rate of formation of nuclei $I^{homo}$ between supercooled erythritol and CG-thickened erythritol (15 wt.%) at 80 °C. **k** Change of $I^{homo}$ of erythritol and thickened erythritol as a function of temperature from 0 K to 391 K and of interfacial energy $\gamma_{sl}$ from 0 mJ·m$^{-2}$ to 140 mJ·m$^{-2}$. **l** Comparison of the decreasing on crystallization duration of the 15 wt.% CG-thickened erythritol in relative to that of pure erythritol. **m** Comparison of the improvement on $\gamma_{sl}$ of erythritol between CG and other common thickeners including PVA, CMC and CMCNa, at a loading of 2 wt.%.

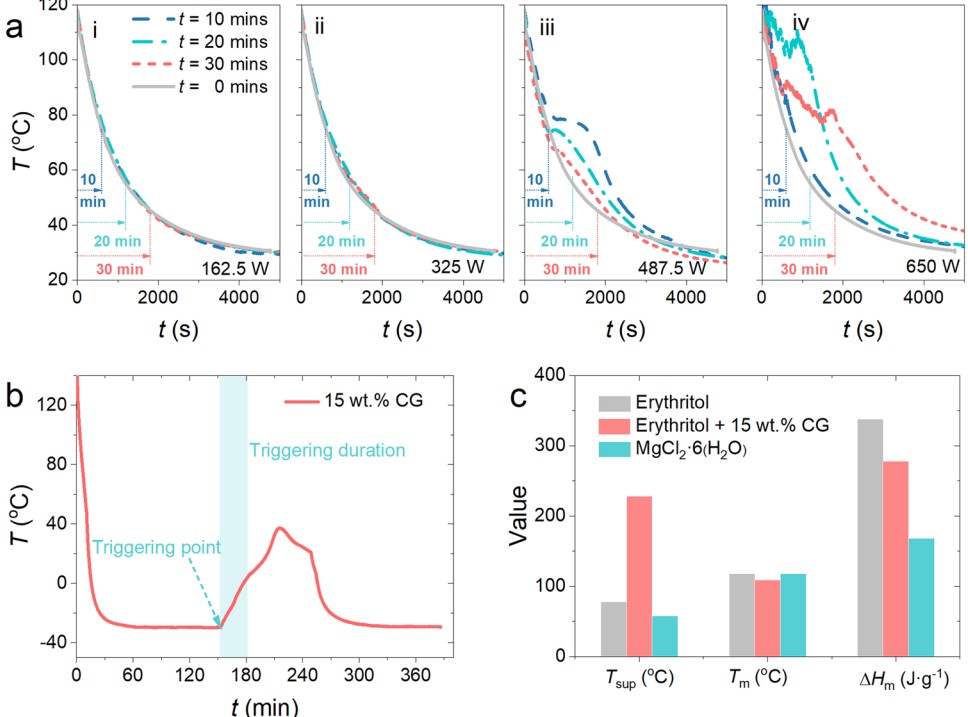

**Fig. 6 | The controllable triggering of crystallization process and the comprehensive evaluation of CG-thickened erythritol. a** The effects of duration and power of ultrasonication on triggering of the 15 wt.% CG-thickened erythritol. **b** Triggering the crystallization at a power of 487.5 W for a duration of 30 min after

the CG-thickened erythritol has been cooled down at a low temperature of −30°C. **c** A comprehensive evaluation of the CG-thickened erythritol in consideration of several aspects including the degree of supercooling, melting point, latent heat of fusion, sustainability, cycling stability, and corrosivity.

## Active triggering of crystallization of CG-thickened erythritol by ultrasonication

The CG-thickened erythritol, at the loading of 15 wt.%, cannot undergo spontaneous crystallization upon cooling, thus requiring external energy to offset the increased energy barriers of crystallization to trigger the discharging process, as indicated in Fig. 5h. In general, seeding, partial cooling, stirring, and ultrasonication are among the widely-used methods to trigger the crystallization of supercooled PCMs. Stirring and ultrasonication both belong to the mechanical disturbance for triggering crystallization. However, the application of stirring is difficult due to the very high viscosity of the thickened erythritol. Partial cooling is not compatible with this scenario because the thickened erythritol can maintain the supercooled state even down to −40°C. Introducing crystal seeds as external nuclei is always a viable option, but it seems impractical for real-world TES applications that require supplementing the seeds for each triggering shot. Therefore, ultrasonication was selected as the preferred method for triggering the thickened erythritol, as it

can be operated in an "on-off" way with ease of tuning the power and duration on demand.

As shown in Fig. 6a, the effectiveness of triggering, the 15 wt.% CG-thickened erythritol, depends on the power and duration of the ultrasonic stimulus applied. At low powers (e.g., <325 W), regardless of the duration (up to 30 min), the crystallization process cannot be activated and the CG-thickened erythritol always remains the supercooled state with lowering the temperature. As shown in Fig. 6a-iii & iv, only when the ultrasonic power surpasses a certain threshold, becoming above 487.5 W in this work, could the crystallization and discharging process be successfully triggered by ultrasonication. Note that our goal was to verify if ultrasonication works for triggering of our new highly-supercooled PCM, so the exact threshold power was not pursued. Crystal nuclei initially form in the vicinity of the ultrasonication probe, followed by growth into complete crystals as the latent heat of the supercooled CG-thickened erythritol is fully released. Visualized observation of the crystallization process triggered by ultrasonication is presented in Supporting information (SI 5).

Similar to the effect of ultrasonication power, the duration of ultrasonication for triggering also seems to have a threshold value. As shown in Fig. 6a, we found that when the duration is less than 10 min, even if the power of ultrasonication reaches 650 W, the CG-thickened erythritol still cannot be triggered to crystallize. Only when the duration of ultrasonication exceeds 10 min, could crystallization be properly triggered. In addition, a longer triggering duration may lead to an undesirable negative effect on crystallization, because the prolonged ultrasonication will break the growing erythritol crystals into smaller fragments that may then serve as new crystal nuclei in the supercooled erythritol.

Instead of the above-triggering tests at the cooling process, we also tried to trigger the 15 wt.% CG-thickened erythritols in a stable supercooled state at a severe cold temperature, mimicking the potential application scenarios of the seasonal TES in a severely cold environment. As plotted in Fig. 6b, we showed that using ultrasonication can still trigger the crystallization process successfully when the sample was cooled down to an ultralow temperature of −30°C using a chiller. Note that −30°C is the target temperature for our seasonal TES applications (see Fig. 1b), showing that the ultrasonication method is compatible with such severe cold weather.

We also proved that the power required to trigger crystallization is not sensitive to the mass of the sample. We tested the triggering of samples having a weight from 20 g to 200 g using a constant power of 487.5 W and a duration of 30 min (see Supporting information (SI 5)), and found that they all successfully crystallized. After being triggered, the crystallization process continues for all the CG-thickened erythritol samples tested, as confirmed by the continuous temperature rising after turning off the ultrasonication in Fig. 6b, until a complete discharge of the latent heat. As a seasonal TES system of practical scales only requires one triggering shot when needed, the ultrasonication method is expected to be energy-efficient.

The active triggering of crystallization by ultrasonication seems to hold the "key" to the "lock" for the well-protected latent heat in the highly supercooled erythritol through thickening. As summarized in Fig. 6c, the CG-thickened erythritol (at the loading of 15 wt.%) has a significant advantage in terms of the supercooling behavior compared to pure erythritol, despite a slight sacrifice in the latent heat. However, the CG-thickened erythritol still has a high latent heat of fusion >200 J·g$^{-1}$, which is greater than other supercooled non-erythritol PCM, e.g., $MgCl_2 \cdot 6(H_2O)$ within the same operating temperature range. In addition, as a pure organic and bio-derived composite PCM, the CG-thickened erythritol offers several advantages compared to other inorganic PCMs, including high environmental friendliness, stable cyclic performance, negligible corrosivity, and so on. In particular, for long-term seasonal storage applications, the ultrastable supercooling of CG-thickened erythritol can greatly lower the thermal insulation cost of practical-scale TES systems.

## Discussion

To enable a highly-supercooled PCM for seasonal latent heat storage in severe cold weather conditions, a strategy of doping a natural food thickener into erythritol is proposed, to further improve its relatively high degree of supercooling ( ~ 60°C) to an unprecedented ultrahigh level of >200°C upon adding 15 wt.% of CG. The CG-thickened erythritol possesses an increased interfacial energy (by ~45%) and a boosted viscosity (by >300 times), thus acting as a high thermodynamic "dam" to prevent any inadvertent loss of the stored latent heat during long-term, seasonal storage periods. The ultrahigh degree of supercooling of CG-thickened erythritol guarantees stable storage of the latent heat in any cold place on Earth. The common ultrasonication method can trigger crystallization of the CG-thickened erythritol during the cooling process, thus enabling a controllable "key" for the timely release of the "locked" latent heat when needed.

In addition to its ultrahigh degree of supercooling that enables ultrastable seasonal storage and little concern for thermal insulation and corrosion, the CG-thickened erythritol is also highly sustainable because of both ingredients, i.e., erythritol and CG, are biomaterials that have long been used in food, pharmaceutical, and chemical industry. We believe that this simple yet effective thickening strategy can be extended to other types of PCM in practical TES scenarios with various temperature ranges. The overall high performance of the CG-thickened erythritol makes it a very promising eco-friendly, mid-temperature PCM for seasonal storage of solar thermal energy.

## Methods

### Preparation of the thickened-erythritol samples

The erythritol used in this work was purchased from Macklin Ltd., and the bio-derived food thickeners, i.e., carrageenan gum (CG), guar gum (GG), and xanthan gum (XG), and other common type thickeners, i.e., polyvinyl alcohol (PVA), carboxymethyl cellulose (CMC) and sodium carboxymethyl cellulose (CMCNa) were purchased from Aladdin Ltd. Other information of the reagents is given in Supplementary Table 3 in Supporting information (SI 1&2). A planetary ball mill (QM-3SP04) was used for grinding and mixing the thickener and erythritol. Then the mixture was heated to melt on a constant temperature heating plate (IKA c-MAG HS 7) at the temperature of 160°C. Finally, cooling was applied to the molten samples at the temperature of 25°C to make them crystalize.

The solidified samples were then subjected to a series of characterizations including chemical and structural characterization like SEM (S-3700N, HITACHI, Japan), XRD (APEXII, Bruker, Germany), FTIR (Vertex 70, Bruker, Germany), isothermal (DSC, NETZSCH DSC 200 F3, Germany) and non-isothermal tests for phase change performance, as well as the thermal stability (TGA, TGA/DSC3 + , Mettler Toledo, Swiss) and rheological tests (MCR 102, Anton Paar, Austria), as shown in Supplementary Fig. 1 in Supporting information (SI 1). Some samples that cannot crystallize spontaneously due to the high loading of thickeners were triggered by ultrasonication. Details regarding this active triggering are discussed in Supporting information (SI 5).

### Non-isothermal tests

All the as-prepared samples were subject to non-isothermal heating and cooling cycles by differential scanning calorimetry (DSC), to measure the important phase change properties such as the melting point, latent heat of fusion, and degree of supercooling. DSC testing was conducted using a sealed alumina crucible with a sample mass of ~15 mg, over the temperature range from −100°C to 150°C. After heating or cooling to the set temperature, the sample was held isothermally for 10 min to ensure complete melting or cooling. The heating and cooling rate was 10 K·min$^{-1}$, and a total of 3 cycles were performed for each sample. The testing was carried out under a high-purity nitrogen atmosphere, with a purge flow rate of 20 ml·min$^{-1}$. The starting temperature of the endothermic peak was taken as the melting point, and the end point of the exothermic peak was identified as the onset point of crystallization. Both temperatures were recorded to calculate the degree of supercooling. The latent heat values of fusion ($\Delta H_m$) and crystallization ($\Delta H_c$) were obtained by integration to calculate the area under the endothermic and exothermic peaks, respectively.

### Isothermal heating/cooling tests

Isothermal heating and cooling tests were conducted based on the temperature history (T-history) method, recording the phase change behaviors of erythritol and the thickened erythritol when being cooled down in a constant-temperature environment. The experimental setup of isothermal heating and cooling tests is shown in Supplementary Fig. 2 in Supporting information (SI 1). First, a 10 g sample was placed in a quartz test tube with an inner diameter of 13 mm and an outer

diameter of 15 mm. Then, the sample was heated up for 30 min on a constant-temperature heating plate at the temperature of 160°C for melting.

After complete melting, the test tube was moved into a temperature-controlled insulation chamber. The temperature of the chamber was controlled by a water bath, and the temperature curve of the sample during the heating and cooling processes was recorded by a T-type thermocouple. The heating-cooling process was repeated twice for each sample.

For observing the supercooling behaviors of erythritol and the thickened erythritol at various isothermal temperatures, the water bath temperatures were set at 90°C, 50°C, and 10°C. The supercooling behavior in extremely low-temperature environments was achieved by liquid nitrogen. As shown in Supplementary Fig. 2, liquid nitrogen was pumped into the insulation chamber to maintain an ultralow temperature (about −140°C). Unlike the precisely controlled temperatures of 90°C, 50°C, and 10°C maintained by the water bath, such ultralow temperature was only controlled within a broader temperature range lower than that of liquid nitrogen (−196°C) due to the flow rate limitations of the liquid nitrogen pump and heat loss from the chamber. In a typical run at a given temperature, all the samples were moved into the chamber simultaneously to experience the same heating/cooling histories for comparison. In addition, the internal temperature of the chamber was recorded using another T-type thermocouple.

## Determination on the rheological behaviors

The rheological behaviors of the samples were characterized using a stress-controlled modular compact rheometer (Anton Paar, MCR102) with precise temperature control up to 400°C, as shown in Supplementary Fig. 3 in Supporting information (SI 1). It used a cone-plate system to ensure that the shear rate and shear stress were the same throughout the flow field. Firstly, the powder of the sample with a mass of about 2 g was spread on the plate of the testing system. The temperature of the rheometer was set to 150°C to melt the sample. The temperature of the testing system was controlled by an electrical heating device with a temperature accuracy of ±0.01°C. After the sample was melted, the cone was slowly lowered until the gap between the cone and the plate reached the set value of 0.1 mm. Then, the insulating cover of the rheometer was manually placed down to completely cover the cone-plate system, reducing temperature fluctuations during the test. The viscosity of the molten sample at 150°C was measured first, and then the viscosity of the sample was measured with a temperature-decreasing gradient of 150–130°C. During the cooling process, the testing system was cooled by air supplied by the air compressor.

## Measurement of the solid-liquid contact angles by sessile drop method

To determine the interfacial energy between the solid and liquid phases of pure erythritol or thickened erythritol, the contact angle between the solid-liquid phases of the same sample must be measured. To minimize the effect of surface roughness, the surface of the solidified sample was carefully polished using a 1200-grit sandpaper. Subsequently, the solid and supercooled erythritol was placed in a thermostatic chamber at the temperature of 80 °C for 40 min. After achieving a stable temperature, a drop of the supercooled sample was dispensed onto the surface of the solid sample. The samples were incubated in the thermostatic chamber to facilitate the complete spreading of the supercooled drop and attain a stable state for measuring the contact angle. Subsequently, the contact angle between the supercooled and solid samples was determined using a contact angle goniometer (POWEREACH), as shown in Supplementary Fig. 11 in Supporting information (SI 3).

It is important to note that once the supercooled sample is dispensed onto the surface of the solid sample, the solid surface could act as a nucleating heterogeneity, leading to the crystallization of the supercooled erythritol before complete spreading. However, for reliable contact angle measurements, the liquid erythritol sample should be well controlled to have a proper temperature for it to maintain the supercooled state without freezing on the solid surface. So the testing temperature needs to be held as high as possible but below the melting point of erythritol to keep simultaneously a supercooled state of the liquid drop as well as the solid surface underneath without melting.

The contact angles of the PVA-, CMC- and CMCNa-thickened erythritol samples were measured by the same sessile drop method, as presented in Supporting information (SI 3.1). The measurements were also done at the temperature of 80°C, and all the samples kept the same thickener loading of 2 wt.%. Here, we tested the contact angle of 2 wt.% CG-thickened erythritol for comparison with other thickeners, as shown in Supplementary Fig. 17 in Supporting information (SI 4).

## Measurement of the solid-liquid interfacial energy by pendant drop method

To obtain the interfacial energy between the solid and liquid phases of the samples, the second step is to estimate the interfacial energy between the liquid sample and air (i.e., the surface tension of the liquid sample) through the pendant drop method.

There is definition of a function of drop shape S, as given by[82],

$$S = \frac{d_s}{d_e} \tag{3}$$

As depicted in Supplementary Fig. 12a in Supporting information (SI 3), using a pendant water drop as an example, $d_e$ is the maximum (equatorial) diameter of the pendant drop and $d_s$ is the diameter of the pendant drop in a selected plane at a distance $d_e$ from the apex of the drop. In this work, the S of erythritol and CG-thickened erythritol was both measured three times to reduce measurement errors, as shown in Supplementary Fig. 12b and c, respectively. The S factors of erythritol (sample #1, #2, #3) and CG-thickened (sample #1, #2, #3) are 0.7166, 0.7096, 0.6166 and 0.5922, 0.6171, 0.6112, respectively.

Based on the value of the S, another function of drop shape H can be calculated by[82]

For S > 0.59 to S = 0.68

$$1/H = (0.31522/S^{2.62435}) - 0.11714S^2 + 0.15756S - 0.05285 \tag{4}$$

For S > 0.68 to S = 0.90

$$1/H = (0.31345/S^{2.64267}) - 0.09155S^2 + 0.14701S - 0.05877 \tag{5}$$

The final gas-liquid interface $\gamma_{lg}$ can be obtained by Eq. (7), and the detail parameters are shown in Supplementary Table 3 in Supporting information (SI 3).

$$\gamma_{lg} = \frac{\Delta\rho g d_e^2}{H} \tag{6}$$

where $\Delta\rho$ is the density difference between gas and liquid.

Based on Eq. (6), we obtained the gas-liquid interfacial energy of $34.31 \pm 2.0$ mJ·m$^{-2}$ for pure erythritol, which is lower than the $52.68 \pm 0.54$ mJ·m$^{-2}$ for the 15 wt.% CG-thickened erythritol. The calculated results of the intermediate parameters and the interfacial energy are given in Supplementary Table 4 in Supporting Information (SI 3).

Due to the high viscosity of the thickened erythritol samples by the other thickeners, the liquid-gas interfacial energy was hard to measure by the pendant drop method. Alternatively, we measured it at a temperature of 80°C by an automatic surface tensiometer (AFES,

FST300M), as shown in Supplementary Fig. 18a in Supporting information (SI 4).

## Active triggering of crystallization

After thickening the erythritol with 15 wt.% CG, the thickened erythritol sample was transferred to the ultrasonic device for triggering tests, as shown in Supplementary Fig. 19 in Supporting information (SI 5). The thermocouple and ultrasonic probe were inserted into the molten sample, and then the temperature of the sample was gradually cooled down to room temperature, forming the supercooled sample. After stabilizing at room temperature for a certain period, the crystallization and discharging process was triggered by activating the ultrasonication. The power and duration of the ultrasonication applied were turned by the controller, which is deemed to be two core factors affecting the successfulness of triggering. In the triggering process, a comparison was made for four different power levels (162.5 W, 325 W, 487.5 W, 650 W) and three triggering durations (10 min, 20 min, 30 min).

## Reporting summary

Further information on research design is available in the Nature Portfolio Reporting Summary linked to this article.

## Data availability

Source data are provided in this paper.

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

## Acknowledgements
This material is based upon work supported by the National Natural Science Foundation of China under Grant No. 52276088, and the Fundamental Research Funds for the Central Universities under Grant No. 2021FZZX001-10.

## Author contributions
S.Y. conceptualized, investigated the project, designed and performed the experiment, and wrote the manuscript. H.S., J.L., and Y.L. performed the experiment. Ö.B. edited the manuscript. L.F. conceptualized, investigated the project, supervised, wrote, reviewed, and edited the manuscript. The manuscript was written through the contributions of all authors. All authors have given approval to the final version of the manuscript.

## Competing interests
The authors declare no competing interests.
