## [Peer Review File · Nature Communications]

Supercooled erythritol for high-performance seasonal thermal energy storageREVIEWER COMMENTS

Reviewer #1 (Remarks to the Author):

The manuscript: Ultrastable supercooled erythritol that can withstand down to -100°C for high-performance seasonal thermal energy storage, presents an interesting and current scientific issue relating with long-term heat storage. The described solution concerns erythritol composites, and the presented results confirm certain aspects of its future application. However, the deeper mechanism of the CG applied on erythritol is lacking. Also the shortcomings have been observed as the following comments:

1. As the manuscript described, the increase in viscosity is why CG maintains the supercooling stability of the erythritol. While many thickeners (such as CMC, PVA, or other gums) can do a similar rheology behavior as the CG-erythritol composite. This could also change the Gibbs free energy of pure erythritol. What is the difference between the chosen thicker and the others? Please specify the deeper mechanism.
2. The author used the glass tube to complete the T-history experiment. I wonder how to do cycles. The glass tube may crack when you reheat the erythritol and its composites. Cracking is very likely to occur due to the stress caused by the thermal expansion of the solid erythritol, whether it is heated by a cavity or an oil bath. Please explain the special heating method.
3. Fig.3: The discharging latent heat of 15% CG adding is almost equal to 10% CG adding. It is unusual. Hope the author can explain in further.
4. What is the success rate when using ultrasonic for triggering? Also, the thermocouple and ultrasonic probe were inserted into the molten erythritol as Fig.S11. The erythritol could be triggered by the probe because of the particle cooling of the steel probe (it connects to surroundings) or the heterogeneous nucleation instead of ultrasonic. Please specify how to rule out these factors.
5. The energy of ultrasonic fluctuations is converted into heat, which in turn may cause local erythritol to overheat or even degrade. The author determined the power of ultrasound to be 487.5W and 650W. Is there any risk of local degradation of erythritol?
6. Continuing from the previous discussion, the author states that a seasonal TES system only requires one triggering shot when needed. As references show, seeding, partial cooling or stirring methods are seen to be more reliable than ultrasonic. Why did the author choose this triggering method? Please specify the reason.

Reviewer #2 (Remarks to the Author):

The paper experimentally investigates the potential of carrageenan-thickened erythritol for seasonal thermal energy storage. Especially, the high and stable supercooling degree of the material might reduce heat losses from the storage, even in severe cold ambient temperature. The paper is very interesting paper but still needs to be significantly improved before it is suitable for publication. The following major and minor comments should be addressed:

Major comments:

1. Fig. 3b: why is latent heat different from charge to discharge? Similarly, the sentence "lower crystallization temperatures result in reduced release of latent heat during the discharging process, attributed to the disparity in heat capacity between the solid and liquid phases of erythritol" should be better explained. Actually, latent heat (a constant thermophysical property for a given material and a given phase transition) does not change. Part of it is simply consumed. During discharge, part of Latent heat (actually $C_{p_liquid} \times (\text{Melt temperature} - \text{Nucleation temperature})$) is used to adiabatically heat the material up to the melt temp. The remaining (if any) energy, which is Latent heat - $C_{p_liquid} \times (\text{Melt temperature} - \text{Nucleation temperature})$, is then available for external use. I suppose the observed decrease is due to the consumed energy.
2. Major thermophysical properties of the materials used should be provided in a table (especially the latent heat and liquid and solid heat capacities for pure erythritol and 15 wt% CG-erythritol). It will help provide some numerical assessment of the amount of energy that can be stored and

recovered (see previous comment).

3. 6%, 11% and 4% "reduction" in latent heat correspond respectively to 5wt.%, 10wt.% and 15wt.% CG-thickened erythritol. Why is it nonlinear?

4. In fig.3(e), cold crystallization seems to augment with subsequent charging phases. please comment on that.

5. Is the power required to trigger crystallization mass-dependent? In other terms, would the power required to trigger a very large quantity of CG- thickened erythritol (used for seasonal energy storage) be quite significant? If so can the author provide an estimation of the needed power versus CG- thickened erythritol mass or volume?

6. Temperature axes in Fig 6 show positive values only. Triggering should be experimented for negative values too (as the compound might be triggered at temperatures down to -100 °C).

Minor comments:

1. The size of some figures should be increased, as they are hardly readable when printed in regular paper format (e.g., Fig 3 (a) and (d)).

2. The English writing of the paper is generally good but it still needs to be thoroughly checked, as many misspellings can be found. Here are a few examples (I am not being exhaustive):

Line 371: "When the addition of CG at... » should read « When adding CG at..."

Line 408: "the process...continue" should read "the process...continues".

Line 220: more "than" 60 days.

3. All variables in the equations embedded in fig. 4a must be defined. Provide reference for those equations.

4. Line 299: did you mean Fig 4(a)?

5. Using γ_{sl} in Fig 4 and γ_{ls} in eq. 1 is confusing (the swapped index makes the storyline hard to follow).

6. Line 339: did you mean γ_{sl} ?

7. Fig. 5. Using symbol γ again as shear rate increases confusion. I recommend using another symbol.

Reviewer name:

Prof. Pascal BIWOLE, Ph.D.

Clermont Auvergne University,

France

RESPONSE TO REVIEWERS' COMMENTS

Reviewer #1 (Remarks to the Author):

The manuscript: *Ultrastable supercooled erythritol that can withstand down to -100°C for high-performance seasonal thermal energy storage*, presents an interesting and current scientific issue relating with long-term heat storage. The described solution concerns erythritol composites, and the presented results confirm certain aspects of its future application. However, the deeper mechanism of the CG applied on erythritol is lacking. Also the shortcomings have been observed as the following comments:

- 1) As the manuscript described, the increase in viscosity is why CG maintains the supercooling stability of the erythritol. While many thickeners (such as CMC, PVA, or other gums) can do a similar rheology behavior as the CG-erythritol composite. This could also change the Gibbs free energy of pure erythritol. What is the difference between the chosen thicker and the others? Please specify the deeper mechanism.

Response#

Thank you for raising this insightful concern and your valuable suggestion. As discussed in **Supplementary information (SI 2)**, two other gum-type thickeners, i.e., Xanthan gum and Guar gum, had already been tested. It was shown that they lead to less remarkable improvement in the degree of supercooling when compared to CG, although both can also increase the viscosity of erythritol. Taking your suggestion, to give a more comprehensive comparison and analysis, we conducted extra experiments on testing the enhancing effects of other types of common thickeners, including polyvinyl alcohol (PVA), carboxymethyl cellulose (CMC) and sodium carboxymethyl cellulose (CMCNa). The new comparative results have been added into **Supplementary information (SI 4)**.

As presented in Fig. R1, a comparative analysis is given on the impact of various thickeners on the supercooling behavior of erythritol at the same loadings. In the non-isothermal tests by DSC, CMC, CMCNa and CG, when the loading >10 wt.%, all result in a significantly high degree of supercooling, except for PVA. The PVA-loaded samples (up to 15 wt.%) can crystallize during the cooling process, as indicated by the small peaks on the exotherms. In addition to the low thickening effect, PVA is also classified as a carcinogenic substance (by Toxic Substances Control Act), so it seems not suitable for application as a thickener for making highly-supercooled erythritol.

Fig. R1 The supercooling behavior and cold crystallization of erythritol thickened by various types of thickeners under non-isothermal test condition.

As shown in Fig. R2, subsequent isothermal test results showed that 15 wt.% PVA-, 15 wt.% CMC- and 15 wt.% CMCNa-thickened erythritol are unable to retain supercooled state at low temperatures below 0°C, resulting in rapid crystallization and discharging upon being cooled down when exposed to the cold environment. While erythritol near the cold tube wall promptly generates crystal nuclei and releases the heat, the portion close to the middle, i.e., near the thermocouple, has not yet been sufficiently cool. Such considerable temperature gradient within the test tube results in the observation that on the temperature-time curve of CMCNa-thickened erythritol it reads a crystallization temperature of 97°C which is higher than that of pure erythritol in the subsequent cooling process.

Fig. R2 The isothermal test results of the thickened erythritol samples with various types of thickeners at the same loading of 15 wt.%.

Due to the extremely outstanding thickening effect of CMC and CMCNa, it was difficult to measure the viscosity of the most concentrated 15 wt.% samples. Therefore, we compared the addition of 2 wt.% with CG. As shown in Fig. R3, it can be confirmed that CMC and CMCNa exhibit much higher thickening effect than CG, while PVA shows a lower thickening effect than CG, at the same 2 wt.% loading. However, why did CMC and CMCNa perform less effective, as compared to CG, in improving the degree of supercooling of erythritol?

Fig. R3 The measured rheological behaviors of the thickened erythritol samples.

Here, as inspired by your question and the editorial concern, we would like to discuss the deeper mechanism for the improvement of supercooling of erythritol. In order to address this concern, such discussion has been added in **Section 2.4**. The crystallization process consists of two steps, as shown in Fig. R4. The first step involves the growth of embryos into nuclei with a critical nucleation radius, followed by the second step where the nuclei develop into complete crystals. The solid-liquid interfacial energy (γ_{sl}) of the PCM plays a critical role in the first step, and viscosity (μ) dominates the second step. Yuan et al. (*Chem. Eng. J.* 2023, 469, 143743) has investigated the effect of viscosity on the growth of nuclei during the second step, and viscosity has been clearly confirmed to have the most significant impact on this step. Therefore, in this work, we paid special attention on the effect of solid-liquid interfacial energy in the first step.

Fig. R4 The two steps of the crystallization process, where the first step involves the growth of embryos into nuclei with a critical nucleation radius, followed by the second step for the nuclei to develop into complete crystals.

In the first step of crystallization, both the critical size of nucleus R_c and rate of nucleus formation I^{homo} are crucial factors dictating the formation of the nuclei. It is widely accepted that the concentration of embryos needs to reach $10^6/\text{m}^3$ in the supercooled liquid to form nuclei (J. A. Dantzig, M. Rappaz, *Solidification Chapter 7*, CRC Press, 2009). However, the significant enlargement of R_c necessitates a higher concentration which becomes up to $10^{13}/\text{m}^3$ of embryos in the CG-thickened erythritol. Consequently, the nucleation process becomes much more challenging. Based on the equation about I^{homo} in Fig. 4(b) of the main text, the relationship between I^{homo} and temperature T and solid-liquid interfacial energy γ_{sl} can be obtained. As a function of temperature, the I^{homo} of erythritol is significantly decreased by 10^{10} times with the addition of CG (see in Fig. R5(a-i)). When the temperature approaches 0 K or the melting point of erythritol (~ 391 K), the drop in I^{homo} becomes much sharper. Compared to pure erythritol, the I^{homo} of the CG-thickened erythritol decreases by more than 10^{160} times at the temperature of 353 K.

As shown in Fig. R5(a-ii), the I^{homo} decreases with increasing γ_{sl} at a constant

temperature. In theory, erythritol cannot crystallize spontaneously at a temperature higher than 100°C (~373 K), but our experimental results showed that erythritol fails to crystallize at a temperature higher than 60°C (~333 K). Because in the supercooled state, erythritol undergoes both crystal growth from embryos to nuclei and embryo decay (J. A. Dantzig, M. Rappaz, *Solidification Chapter 7*, CRC Press, 2009). So, it requires a minimum I^{homo} to cover the decay rate of embryos. It can be considered that the I^{homo} of erythritol at 60°C (~333 K), and with γ_{sl} of 33.98 mJ/m², is the minimum rate at which it can crystallize spontaneously.

With the addition of CG, the theoretical minimum supercooling temperature can be lowered from 60°C to -70°C. Furthermore, due to the thickening effect of CG, the tolerable degree of supercooling of erythritol can be further improved. The enlargement of R_c results in a more significant increase in the time needed to form complete nuclei. As depicted in Fig. R5(b), the duration of the nucleation process is extended by a factor of at least 10¹⁰. Therefore, a significant improvement in the degree of supercooling of erythritol was ultimately achieved through such dual barriers of the increase in R_c and the decrease in I^{homo} , leading to the ultrastable supercooling behavior and ultrahigh degree of supercooling of the CG-thickened erythritol.

Fig. R5 The change of I^{homo} of erythritol and thickened erythritol as a function of (a-i) temperature from 0 K to 391 K, and as a function of (a-ii) interfacial energy γ_{sl} from 0 mJ/m² to 140 mJ/m². (b) The temperature-dependent variation of the crystallization duration of the 15 wt.% CG-thickened erythritol in relative to that of pure erythritol.

As mentioned, thickeners of PVA, CMC and CMCNa were found to have worse performance in improving in the supercooling behavior of erythritol than CG. As shown in Fig. R6, despite CMC and CMCNa exhibiting a more pronounced increase in viscosity than that of CG, their lower effectiveness in enhancing the solid-liquid interfacial energy leads to an easier formation of crystal nuclei in the

first step of crystallization, as shown in Fig. R5. As a result, CG stands out as the best, among all the tested thickeners, in stabilizing the supercooling behavior of erythritol.

Fig. R6 Comparison of the improvement on γ_{sl} of erythritol between Carrageenan (CG), polyvinyl alcohol (PVA), carboxymethyl fiber (CMC) and sodium carboxymethyl cellulose (CMCNa), at the same loading of 2 wt.%.

- The author used the glass tube to complete the T-history experiment. I wonder how to do cycles. The glass tube may crack when you reheat the erythritol and its composites. Cracking is very likely to occur due to the stress caused by the thermal expansion of the solid erythritol, whether it is heated by a cavity or an oil bath. Please explain the special heating method.

Response#

Thank you for raising this concern. It is true that pure erythritol has the potential to break the test tube when it is being heated and melted because of its high thermal expansion coefficient, as shown in Fig. R7.

Fig. R7 The thermal expansion coefficient of erythritol and the 15 wt.% CG-thickened erythritol.

However, when erythritol is thickened by 15 wt.% of CG, the expansion coefficient decreases by ~20% (the slope represents the thermal expansion coefficient), thus lowering the risk of breaking the test tube. To avoid tube breakage, we actually did the heating and melting very carefully using a high heating temperature (at a wall superheat of 40°C). The larger temperature difference ensures that the erythritol or thickened-erythritol near the inner wall of the test tube melts quickly before the temperature in the middle of the test tube rises, so as to provide an expansion space, through squeezing between the inner wall of the test tube and the remaining solid PCM, for accommodating the volume expansion during melting. In this way, we did not observe any cracking or breaking accidents during our heating/cooling runs

- 3) Fig.3: The discharging latent heat of 15% CG adding is almost equal to 10% CG adding. It is unusual. Hope the author can explain in further.

Response#

Thank you for raising this concern, the discussion about this observation has been added in **Section 2.2** of the main text and **Supporting information (SI 1)**.

In order to provide a solid thermodynamic elucidation on this unusual observation, here we would like to start from the basic calculation of the latent heat of a PCM. In general, the latent heat refers to the difference in enthalpy before and after the phase transition of a material, such as melting or crystallization. Since the nominal melting point of a pure substance is fixed, it might give people the impression that the latent heat is a constant value, which is not always true.

During the charging process, the latent heat of fusion ΔH_m can be estimated by

$$(1)$$

where H_{Ti} is the enthalpy of the PCM at the temperature of T_i , e.g., T_1 is the temperature at the end of the melting process, and T_2 is the temperature at the beginning of the melting process. The PCM is in solid phase when the temperature is lower than T_2 , and in liquid phase when temperature is higher than T_1 .

Similarly, during the charging process, the latent heat of crystallization ΔH_c can be given by

$$(2)$$

where T_3 is the temperature at the beginning of the crystallization process, T_4 is the temperature at the end of the crystallization process.

Due to the high degree of supercooling of erythritol, and the even higher degrees of supercooling of the CG-thickened erythritol, the temperature of cold

crystallization T_{cc} becomes much lower than the melting point T_m . As shown in Fig. R8(a), the specific heat capacity value of erythritol in the supercooled liquid phase is up to 2.6 J/(gK), while in the solid phase it is less than 1.6 J/(gK). Such a marked difference of the specific heat capacity between the solid and supercooled liquid phases, at the same temperature, leads to such unusual observation that the latent heat of crystallization, which is totally from cold crystallization, of the CG-thickened erythritol at high loadings increases with raising the temperature of cold crystallization. The difference in the latent heat discharged at different temperatures can be intuitively expressed by the shaded area in Fig. R8(a).

Fig. R8 The specific heat capacity and enthalpy of supercooled PCM. (a) The temperature-dependent specific heat capacity of erythritol in both solid and supercooled liquid phases. (b) The change of enthalpy upon the transition of melting and crystallization.

This reasoning can be better elucidated as the T - H diagram depicted in Fig. R8(b). At the melting temperature, the solid erythritol, or any other highly supercooled PCM, absorbs heat and melts into a molten liquid. During the discharging process, the molten erythritol enters a supercooled state upon being cooled down, due to the stable supercooling behavior. Assuming two cold crystallization temperatures of $T_{cc,1}$ and $T_{cc,2}$, with $T_{cc,1}$ being the lower one, the $\Delta H_{c,1}$ is clearly less than $\Delta H_{c,2}$ because the enthalpy of the supercooled liquid phase decreases at a higher rate than that of the solid phase due to their difference in specific heat capacity.

Note that the latent heat values plotted in Fig. 3 of the main text were obtained by averaging over the several consecutive measuring cycles, so the increasing trend of the latent heat of crystallization of the 15 wt.% CG-thickened erythritol becomes larger as the cycling proceeds, as given in **Table R1**. The ΔH_c of the 15 wt.% sample can even become greater than that of the 10 wt.% sample, so their averaged values are found to be close.

Table R1 The temperature of cold crystallization (T_{cc}) and latent heat of crystallization (ΔH_c) of CG-thickened erythritol at high loadings.

Number of cycles	10 wt.%		15 wt.%	
	T_{cc} (°C)	ΔH_c (J/g)	T_{cc} (°C)	ΔH_c (J/g)
1	-3.7	138.8	4.8	118.8
2	-2.0	144.9	2.7	115.0
3	19.2	155.1	5.2	122.2
4	21.4	156.3	10.9	135.7
5	23.0	154.8	18.9	155.8
6	23.6	152.0	23.6	156.6
7	23.6	157.2	28.0	158.9
8	24.2	158.3	31.6	161.3
9	24.0	156.6	31.7	158.9
10	23.9	153.1	32.1	162.3
11	23.2	154.0	30.7	167.9
12	23.5	156.0	31.6	161.2
13	23.5	153.8	32.5	160.7
14	23.8	152.7	32.8	157.1
15	23.8	152.2	33.0	162.0

- 4) What is the success rate when using ultrasonic for triggering? Also, the thermocouple and ultrasonic probe were inserted into the molten erythritol as Fig.S11. The erythritol could be triggered by the probe because of the particle cooling of the steel probe (it connects to surroundings) or the heterogeneous nucleation instead of ultrasonic. Please specify how to rule out these factors.

Response#

Thank you for raising this concern. In our observation, the crystallization process can be triggered successfully by ultrasonication at every time. Moreover, we have eliminated your speculation about the possibility of the steel probe to trigger the crystallization. As depicted in Fig. R9, low-power ultrasonication was unable to trigger the crystallization process in highly supercooled erythritol, so the likelihood of the probe for triggering is very low. To further confirm this point, we conducted an experimental verification, as illustrated in Fig. R10, where the probe was inserted into the highly supercooled erythritol at room temperature for 12 hours, and no crystallization phenomenon was observed at all.

Fig. R9 The effects of the steel ultrasonication probe on triggering.

- 5) The energy of ultrasonic fluctuations is converted into heat, which in turn may cause local erythritol to overheat or even degrade. The author determined the power of ultrasound to be 487.5W and 650W. Is there any risk of local degradation of erythritol?

Response#

Thank you for raising this concern. Alcohol-based PCMs typically produce acids or aldehydes during thermal decomposition. Consequently, we chose the PCM sample in the vicinity to the probe and employed Fourier-transform infrared spectroscopy (FTIR) to assess the emergence of new functional groups. The findings demonstrated the persistence of a stable chemical structure in the erythritol adjacent to the probe. This resilience primarily stems from the efficient dissipation of heat by the highly supercooled erythritol. Furthermore, during our triggering test, the ultrasound was ceased when nucleation begins within the PCM, to prevent oxidation of the material surrounding the probe.

Fig. R10 FTIR curves on the 15 wt.% CG-thickened erythritol before and after being triggered.

- 6) Continuing from the previous discussion, the author states that a seasonal TES system only requires one triggering shot when needed. As references show, seeding, partial cooling or stirring methods are seen to be more reliable than ultrasonic. Why did the author choose this triggering method? Please specify the reason.

Response#

Thank you for raising this concern. After being triggered, the crystallization process continues for all the CG-thickened erythritol samples tested, as confirmed by the continuous temperature rising after turning off the ultrasonication in **Fig. R11**, until a complete discharge of the latent heat. It means that the crystallization can continue by itself after being triggered. So in practical application of thermal storage equipment, as long as there is a point inside the erythritol that is triggered to crystallize, the crystallization can proceed in a “chain-reacting” manner until the latent heat of erythritol is completely released.

Fig. R11 The T -history curve during a typical crystallization process of the 15 wt.% CG-thickened erythritol triggered by ultrasonication.

In addition, as you mentioned, seeding, partial cooling, and stirring have all been widely used for triggering crystallization of PCMs. Among them, the stirring method bears resemblance to ultrasonic triggering, as both rely on an external mechanical energy to disrupt the equilibrium of the supercooled PCM to initiate the crystallization process. However, the use of the stirring method poses a great challenge in this work due to the ultrahigh viscosity of the thickened erythritol, necessitating significant shear stresses. Furthermore, once triggered, the gradual solidification of the PCM impedes the stirring process.

The partial cooling method involves the utilization of localized low temperatures to trigger the formation of crystal nuclei in specific areas within the PCM, thereby facilitating crystallization. However, achieving crystallization through local cooling is incompatible with the case of thickened erythritol, given

its ability to maintain a supercooled state under extremely low temperatures even down to below -30°C .

The only alternative method that seems to work in principle is the introduction of seeds, serving as external nuclei to trigger crystallization. However, in practical applications, PCMs are typically enclosed within thermal storage devices (e.g., heat exchangers), making it impractical to incorporate seed crystals by opening the device before each discharging process.

Therefore, in this work, ultrasonication was selected as the preferred method for triggering the thickened erythritol. This method does not require the thermal storage device to be opened upon each request of heat release; rather, it can be easily operated by turning on or off the switch and the stimulus power and duration are also under control.

Reviewer #2 (Remarks to the Author):

The paper experimentally investigates the potential of carrageenan-thickened erythritol for seasonal thermal energy storage. Especially, the high and stable supercooling degree of the material might reduce heat losses from the storage, even in severe cold ambient temperature. The paper is very interesting paper but still needs to be significantly improved before it is suitable for publication. The following major and minor comments should be addressed:

Major comments:

- 1) Fig. 3b: why is latent heat different from charge to discharge? Similarly, the sentence "lower crystallization temperatures result in reduced release of latent heat during the discharging process, attributed to the disparity in heat capacity between the solid and liquid phases of erythritol" should be better explained. Actually, latent heat (a constant thermophysical property for a given material and a given phase transition) does not change. Part of it is simply consumed. During discharge, part of Latent heat (actually $C_{p_liquid} \times (\text{Melt temperature} - \text{Nucleation temperature})$) is used to adiabatically heat the material up to the melt temp. The remaining (if any) energy, which is $\text{Latent heat} - C_{p_liquid} \times (\text{Melt temperature} - \text{Nucleation temperature})$, is then available for external use. I suppose the observed decrease is due to the consumed energy.

Response#

Thank you for raising this concern, the discussion about this issue has been added in **Section 2.2** of the main text. First, we agree on your explanation on the difference in the latent heat from charging to discharging. It is indeed partially due to the inequality between the melting point and crystallization point of a substance. As the presence of the supercooling effect of a PCM like erythritol, the crystallization point is always lower than the melting point, so that part of the heat between this temperature gap is discharged in the form of sensible heat. In addition, for polymeric materials, partial crystallization may happen that can also lead to a lower latent heat of crystallization than the latent heat of fusion, and the extent of partial crystallization is often quantified by crystallinity.

Secondly, we would to consolidate the discussion regarding our statement of "lower crystallization temperatures result in reduced release of latent heat during the discharging process, attributed to the disparity in heat capacity between the solid and liquid phases of erythritol." This concern is related to the **Comment #3 of the other reviewer**. Here, some of the relevant response, with **Fig. R8**, is presented below again.

Due to the high degree of supercooling of erythritol, and the even higher degrees of supercooling of the CG-thickened erythritol, the temperature of cold

crystallization T_{cc} becomes much lower than the melting point T_m . As shown in Fig. R8(a), the specific heat capacity value of erythritol in the supercooled liquid state is up to 2.6 J/(gK), while in the solid state it is less than 1.6 J/(gK). Such a marked difference of the specific heat capacity between the solid and supercooled liquid phases, at the same temperature, leads to an unusual observation that the latent heat of crystallization, which is totally from cold crystallization, of the CG-thickened erythritol at high loadings increases with raising the temperature of cold crystallization. The difference in the latent heat discharged at different temperature can be intuitively expressed by the shaded area in Fig. R8(a).

Fig. R8 The specific heat capacity and enthalpy of supercooled PCM. (a) The temperature-dependent specific heat capacity of erythritol in both solid and supercooled liquid phases. (b) The change of enthalpy upon the transition of melting and crystallization.

This reasoning can be better elucidated as the T - H diagram depicted in Fig. R8(b). At the melting point, the solid erythritol, or any other highly supercooled PCM, absorbs heat and melts into a molten liquid. During the discharging process, the molten erythritol enters a supercooled state upon being cooled down, due to the stable supercooling behavior. Assuming two cold crystallization temperatures of $T_{cc,1}$ and $T_{cc,2}$, with $T_{cc,1}$ being the lower one, the $\Delta H_{c,1}$ is clearly less than $\Delta H_{c,2}$ because the enthalpy of the supercooled liquid phase decreases at a higher rate than that of the solid phase due to their difference in specific heat capacity.

- 2) Major thermophysical properties of the materials used should be provided in a table (especially the latent heat and liquid and solid heat capacities for pure erythritol and 15 wt% CG-erythritol). It will help provide some numerical assessment of the amount of energy that can be stored and recovered (see previous comment).

Response#

Thank you for your kind suggestion. We strongly agree that the major

thermophysical properties of the materials are important for numerical assessment in future, and such a table (see **Table R2** below) has been added into **Supplementary information (SI 1)**.

Table R2 The major thermophysical properties of erythritol and the CG-thickened erythritol samples.

	pure erythritol	10 wt.%	15 wt.%
T_m (°C)	~118.1±0.7	107.0±1.3	104.7±2.2
ρ (solid at 20°C) (kg/m ³)	1412±39	1384±27	1361±13
ρ (liquid at 150°C) (kg/m ³)	1338±14	1301±8	1250±36
$C_{p,s}$ (solid at 20°C) (J/gK)	1.38±0.11	1.32±0.09	1.30±0.13
$C_{p,l}$ (liquid at 120°C) (J/gK)	2.97±0.17	2.84±0.24	2.74±0.11
ΔH_m (J/g)	335±2.2	279.1±1.1	259.3±15.5

- 3) 6%, 11% and 4% "reduction" in latent heat correspond respectively to 5wt.%, 10wt.% and 15wt.% CG-thickened erythritol. Why is it nonlinear?

Response#

Thank you for raising this concern, which is highly related to your **Comment #1** and the **Comment #3 of the other Reviewer**. The discussion about this observation has also been added in **Section 2.2** of the main text.

As discussed in the main text, for the samples with the two high CG loadings (10 wt.% and 15 wt.%), the latent heat of crystallization is totally composed of cold crystallization discharged upon subsequent heating, rather than during normal cooling process. This nonlinear variation is related to the difference in cold crystallization behavior upon consecutive charging/discharging cycles between the 10 wt.% and 15 wt.% samples. On the charging curves for both the 10 wt.% and 15 wt.% samples, the peak representing cold crystallization keeps shifting to the right, i.e., the temperature of cold crystallization (T_{cc}) becomes higher, as the cycling proceeds, and the extent of shifting for the 15 wt.% case is more remarkable, as clearly shown by the variations of both T_{cc} and ΔH_c over the 15 cycles, as given in **Table R1** (duplicated below again for ease of reading). The greater increase of ΔH_c of the 15 wt.% sample upon cycling makes it to surpass the value of the 10 wt.% sample after the 8th cycle, leading to the very close averaged ΔH_c values (over the several cycles) between these two cases.

Therefore, the origin for this unusual increase of ΔH_c of the 15 wt.% sample with cycling is attributed to the discrepancy of the specific heat capacity between the solid (crystallized) and supercool liquid phases of erythritol (and erythritol-

based composite PCM). As shown in **Fig. R8(a)**, the specific heat capacity have two different values over the temperature range where the ultrastable supercooled state can be held. At the same temperature in this range, the specific heat capacity of erythritol in the supercooled liquid phase is up to 2.6 J/gK, which is much greater than the ~1.6 J/gK in the solid phase. For example, compared to the lower temperature T_1 , the cold crystallization occurs at T_2 will be able to release more heat, as represented by the shaded area marked in **Fig. R8(b)**. So, this fact leads to the relatively high latent heat of crystallization of the 15 wt.% sample, making the reduction in latent heat of crystallization becomes nonlinear at higher CG loadings.

Table R1 The temperature of cold crystallization (T_{cc}) and latent heat of crystallization (ΔH_c) of CG-thickened erythritol at high loadings.

Number of cycles	10 wt.%		15 wt.%	
	T_{cc} (°C)	ΔH_c (J/g)	T_{cc} (°C)	ΔH_c (J/g)
1	-3.7	138.8	4.8	118.8
2	-2.0	144.9	2.7	115.0
3	19.2	155.1	5.2	122.2
4	21.4	156.3	10.9	135.7
5	23.0	154.8	18.9	155.8
6	23.6	152.0	23.6	156.6
7	23.6	157.2	28.0	158.9
8	24.2	158.3	31.6	161.3
9	24.0	156.6	31.7	158.9
10	23.9	153.1	32.1	162.3
11	23.2	154.0	30.7	167.9
12	23.5	156.0	31.6	161.2
13	23.5	153.8	32.5	160.7
14	23.8	152.7	32.8	157.1
15	23.8	152.2	33.0	162.0

- 4) In fig.3(e), cold crystallization seems to augment with subsequent charging phases. please comment on that.

Response#

Thank you for raising this concern. The discussion about this issue has been added in Section 2.2 of the manuscript. Cold crystallization is a common phenomenon observed in sugar alcohols. It occurs when the temperature decreases to a point where the intrinsic power of crystallization potential of the sugar alcohol becomes insufficient to facilitate molecular diffusion and crystallization. This crystallization process can only occur when the temperature rises upon the subsequent heating, rather than being cooled down.

For the highly thickened-erythritol (10 wt.% and 15 wt.%), as given in **Table R1**, the increase in viscosity with the number of heating/cooling cycles leads to the increase of the temperature of cold crystallization. As shown in **Fig. R12**, the viscosity of the 15 wt.% CG-thickened erythritol become larger with increasing the number of cycles, which further intensifies the cold crystallization phenomenon. As the viscosity decreases with increasing the temperature, a higher temperature is required for lowering the diffusion resistance of the thickened erythritol molecules, thus gradually raising the cold crystallization temperature. Therefore, compared to the 10 wt.% sample, the remarkable cycling-induced viscosity variation of the 15 wt.% sample leads to the gradual increase in the temperature of cold crystallization.

Fig. R12 The change in viscosity with the number of cycles for the case of (a) 10 wt.% CG and (b) 15 wt.% CG.

- 5) Is the power required to trigger crystallization mass-dependent? In other terms, would the power required to trigger a very large quantity of CG- thickened erythritol (used for seasonal energy storage) be quite significant? If so can the author provide an estimation of the needed power versus CG- thickened erythritol mass or volume?

Response#

Thank you for raising this concern. The triggering effectiveness by ultrasonication was found to be insensitive to the mass of the samples. As shown in Fig. R13, we tried to trigger the 15 wt.% samples of different weights, i.e., 20 g, 100 g, and 200 g, by ultrasonication at a constant power and duration, and found that they all successfully crystallized.

Fig. R13 Triggering the crystallization of CG-thickened erythritol at different weights using the same power and duration.

Once crystallization is initiated, the process propagates continuously. The contact region between a formed crystal and the supercooled liquid acts as a nucleus, which can further trigger the crystallization of the surrounding supercooled erythritol in a “chain-reacting” manner. As the phase interface transitions to the supercooled state, the latent heat released during crystallization transfers to the supercooled erythritol. In practical applications, multiple triggering points can be rationally arranged to induce simultaneous localized crystallization from various regions in a single large-scale system, thereby enhancing the power of heat discharging.

- 6) Temperature axes in Fig 6 show positive values only. Triggering should be experimented for negative values too (as the compound might be triggered at temperatures down to $-100\text{ }^{\circ}\text{C}$).

Response#

Thank you for raising this concern. The discussion of the triggering at lower temperature environment has been added in **Section 2.5** in the main text and **Supporting information (SI 5)**. We tried to trigger the 15 wt.% CG-thickened erythritol at a much lower temperature of -30°C using a chiller, mimicking the potential application scenarios of the seasonal TES in cold environment.

Fig. R14 The T -history curve during triggering of the 15 wt.% CG-thickened erythritol by ultrasonication at a low temperature of -30°C .

As shown in **Fig. R14**, we found that using ultrasonication, at the power of 487.5 W and with a duration of 30 min, can successfully trigger the crystallization process at -30°C . Note that -30°C is the target temperature for our seasonal TES applications (see Fig. 1(b) in the main text), showing that the ultrasonication method is compatible with severe cold weather. Moreover, the temperature curve keeps rising to reach a maximum value about 40°C after turning off the ultrasonication, which indicates the occurrence of continuous spontaneous crystallization (and continuous heat discharging) after the sample was triggered from a low-temperature starting point.

7) Minor comments:

1. The size of some figures should be increased, as they are hardly readable when printed in regular paper format (e.g., Fig 3 (a) and (d)).

Response#

Thank you for your suggestion. The resolution ratio of the Figures has been increased for better readability.

2. The English writing of the paper is generally good but it still needs to be thoroughly checked, as many misspellings can be found. Here are a few examples (I am not being exhaustive):

Line 371: “When the addition of CG at... » should read « When adding CG at...”

Line 408: "the process...continue" should read "the process...continues".

Line 220: more "than" 60 days.

Response#

Thank you for pointing out the language problems. Misspelling and grammar errors have been corrected throughout the manuscript, to the best of our knowledge.

2. All variables in the equations embedded in fig. 4a must be defined. Provide reference for those equations.

Response#

Thank you for raising this concern. The variables in Fig. 4a have now been defined, and the reference for the equations has been added.

3. Line 299: did you mean Fig 4(a)?

Response#

Thank you for pointing out this typo, which has been corrected.

4. Using γ_{sl} in Fig 4 and γ_{ls} in eq. 1 is confusing (the swapped index makes the storyline hard to follow).

Response#

Thank you for your suggestion. The expression for solid-liquid interfacial energy has been unified as " γ_{sl} ".

5. Line 339: did you mean γ_{sl} ?

Response#

Thank you for raising this concern. The Greek letter gamma was not properly present at several places in our original manuscript file, and this issue has been resolved in this revised manuscript.

6. Fig. 5. Using symbol gamma again as shear rate increases confusion. I recommend using another symbol.

Response#

Thank you for your suggestion. The widely accepted symbol for representing shear rate is actually gamma dot, by the fluid mechanics community. So, in the revised manuscript, we changed the shear rate symbol to gamma dot, which also avoids confusion with gamma for the interfacial energy.

REVIEWERS' COMMENTS

Reviewer #1 (Remarks to the Author):

The authors responded to the comments, the manuscript can be accepted for publication after a final revision. A minor suggestion pointed out as follows:

Expanding the application using the ultrastable supercooling erythritol in Introduction.

Reviewer #2 (Remarks to the Author):

The paper has been significantly improved. I consider it suitable for publication.

RESPONSE TO REVIEWERS' COMMENTS

Reviewer #1 (Remarks to the Author):

The authors responded to the comments, the manuscript can be accepted for publication after a final revision. A minor suggestion pointed out as follows:

Expanding the application using the ultrastable supercooling erythritol in Introduction.

Response

Thank you for your comment and suggestion. We have added one sentence in the Introduction section (at the end of the 2nd paragraph on page 6) to point out other possible application scenarios of the ultrastable supercooling PCM.

Reviewer #2 (Remarks to the Author):

The paper has been significantly improved. I consider it suitable for publication.

Response

No changes/corrections are required.